# Methane Emission Characteristics of Naturally Ventilated Cattle Buildings

**Sabrina Hempel [1],\*** , **Diliara Willink [1]**, **David Janke [1]** , **Christian Ammon [1]** , **Barbara Amon [2,3]** and **Thomas Amon [1,4]**

[1] Department Livestock Engineering, Leibniz Institute for Agricultural Engineering and Bioeconomy (ATB), Max-Eyth-Allee 100, 14469 Potsdam, Germany; dwillink@atb-potsdam.de (D.W.); djanke@atb-potsdam.de (D.J.); cammon@atb-potsdam.de (C.A.); tamon@atb-potsdam.de (T.A.)

[2] Department Technology Assessment and Substance Cycles, Leibniz Institute for Agricultural Engineering and Bioeconomy (ATB), Max-Eyth-Allee 100, 14469 Potsdam, Germany; bamon@atb-potsdam.de

[3] Faculty of Civil Engineering, Architecture and Environmental Engineering, University of Zielona Góra, Licealna 9/9, 65-417 Zielona Góra, Poland

[4] Department of Veterinary Medicine, Institute of Animal Hygiene and Environmental Health, Free University Berlin (FUB), Robert-von-Ostertag-Str. 7-13, 14163 Berlin, Germany

\* Correspondence: shempel@atb-potsdam.de

**Abstract:** The mandate to limit global temperature rise calls for a reliable quantification of gaseous pollutant emissions as a basis for effective mitigation. Methane emissions from ruminant fermentation are of particular relevance in the context of greenhouse gas mitigation. The emission dynamics are so far insufficiently understood. We analyzed hourly methane emission data collected during contrasting seasons from two naturally ventilated dairy cattle buildings with concrete floor and performed a second order polynomial regression. We found a parabolic temperature dependence of the methane emissions irrespective of the measurement site and setup. The position of the parabola vertex varied when considering different hours of the day. The circadian rhythm of methane emissions was represented by the pattern of the fitted values of the constant term of the polynomial and could be well explained by feeding management and air flow conditions. We found barn specific emission minima at ambient temperatures around 10 °C to 15 °C. As this identified temperature optimum coincides with the welfare temperature of dairy cows, we concluded that temperature regulation of dairy cow buildings with concrete floor should be considered and further investigated as an emission mitigation measure. Our results further indicated that empirical modeling of methane emissions from the considered type of buildings with a second order polynomial for the independent variable air temperature can increase the accuracy of predicted long-term emission values for regions with pronounced seasonal temperature fluctuations.

**Keywords:** livestock; greenhouse gas; temperature dependency; barn-specific pattern

## 1. Introduction

Global economic development and population growth resulted in a continuous increase in food demand in the past. This trend is likely to continue in the future [1]. A substantial part of the demand is covered these days by animal-based products, as livestock is capable of converting protein sources that are non edible for humans into high value protein [2]. Livestock production, however, is known to be a key source of air pollution which affects Earth's radiation budget by modifications in greenhouse gas concentrations, in the particulate matter formation and in the plant coverage due to nitrogen deposition. The latest goals to limit global temperature rise well below 2 °C above the pre-industrial level call for a

reliable quantification of such pollutant emissions in all economic sectors, including agriculture [3–5]. In the context of livestock husbandry, in the past decade, mainly the emission dynamics of ammonia have been intensively modeled [6–8]. Other crucial substances that are frequently monitored are greenhouse gases, such as carbon dioxide, nitrous oxide or methane, which are directly affecting Earth's radiation budget [9–14]. Among those, carbon dioxide and methane are special as, in contrast to ammonia or nitrous oxide, they can be produced partly from the manure and partly directly from the livestock (particularly from ruminants).

Carbon dioxide is typically used as a natural tracer to evaluated the air exchange rate of naturally ventilated animal houses in cases where the carbon dioxide emission from the manure is negligible (e.g., barns with concrete floor and frequent cleaning) [12,15–19]. There have been multiple studies around the turn of the millennium that linked the carbon dioxide production per animal to the heat production at a reference ambient temperature (cf. review paper on carbon dioxide production in animal houses [20]). The production term has been found to be specific for animal categories/species (e.g., cattle or poultry) and dependent on the body mass and the feed intake. Furthermore, carbon dioxide showed a daily rhythm that is closely related to the animal activity (cf. also [21]).

Particularly then considering ruminant livestock, methane emissions are highly relevant and there is evidence that those emissions can show similar dynamics as the carbon dioxide [22,23]. A significant formation of both gases from manure is feasible for housing systems where the liquid manure is stored under the slatted floor in the barn at a temperature of more than $15\,^\circ$C. In the case of manure scraper systems, considerably less formation of methane and carbon dioxide from the manure inside the barn is expected. On the other hand, in the case of cows considerable emissions of approximately 6000 L carbon dioxide and 500 L methane per day and animal (the latter corresponds to around $10\,\mathrm{g\,LU^{-1}\,h^{-1}}$) are expected directly from the animals [24].

The emissions of both gases have been found to be correlated with productivity (e.g., milk yield in the case of cattle) and with feed composition and feeding time [23]. However, the ratio between methane and carbon dioxide emissions depends on a wide range of factors and will not be constant. There are crucial differences between carbon dioxide and methane that may lead to different temporal dynamics. For example, a substantial part of the methane emission is associated with fermentation activity in the rumen (about 90% are released by eructation, about 2% by breath), while the carbon dioxide release is mainly associated with respiratory activity.

While mechanistic models for the emission relevant processes in the rumen exist, the upscaling to the barn level is not trivial and was not in the focus of empirical studies so far. In consequence, the emission dynamics of methane inside barns are insufficiently understood. A preceding analysis of methane emissions from a naturally ventilated barn based on long-term measurements at eight indoor and four outdoor sample points with a photo-acoustic analyzer suggested that there is an effect of climatic stress (i.e., suboptimal thermal conditions) on the emission dynamics yielding a parabolic temperature dependence [14].

The present study aims to investigate whether the emission dynamics of methane, that have been described in this former study, can be generalized to other measurements sites and setups. To that end, we considered hourly methane emission values and outdoor temperatures from two naturally ventilated dairy cattle buildings in Germany over approximately ten and five months, respectively. We hypothesized that: (1) Methane emissions from naturally ventilated dairy barns with concrete floor show the parabolic dependence on the air temperature independent of the gas sampling and averaging strategy. (2) The actual shape of the parabola varies in the course of the day, where the rhythm of change and the vertex of the parabola are barn specific. With our study, we want to improve the understanding of methane emission processes from naturally ventilated dairy cattle barns as a basis for the design of efficient and reliable mitigation measures.

## 2. Material and Methods

### *2.1. Data Collection*

The data for this study were collected from two locations in Germany almost continuously over more than five months capturing warm, transition and cold periods. The location and the procedure of data acquisition in both cases are described in the following subsections.

#### 2.1.1. Measurement Sites

We considered two cattle buildings, a larger one (in the following denoted as farm A) and a smaller one (in the following denoted as farm B). The location and orientation of the buildings on the two farms is sketched in Figure 1.

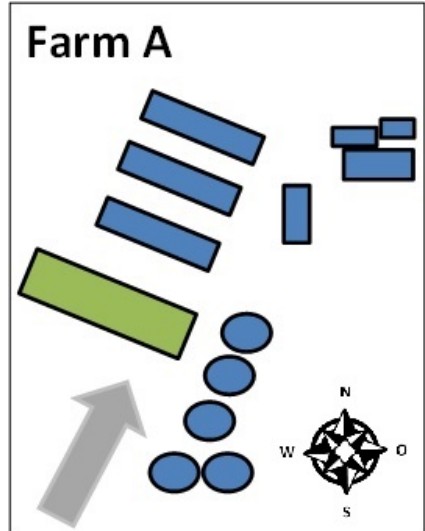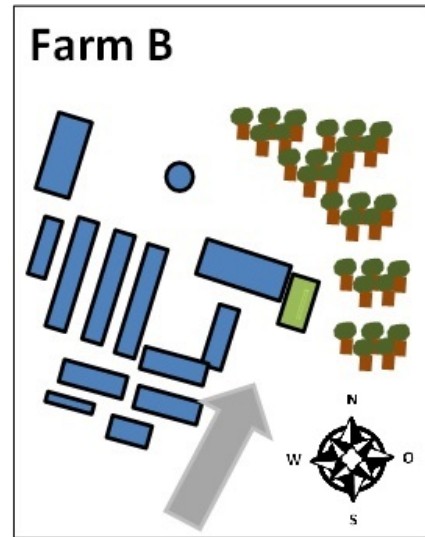

**Figure 1.** Schematic map of the two farm layouts. Green squares indicate the measured barns, blue objects indicate surrounding buildings and storage tanks. Farm B is partly surrounded by forest in addition. The grey arrows indicate the main prevailing wind direction.

Farm A: The first data set originated from a naturally ventilated dairy building located in Mecklenburg-Western Pomerania, north-east Germany (approximately 217 km north-west of Berlin, 42 m above sea level) [25]. The building's total floor area had the dimensions 96.15 m × 34.20 m. The roof has an open ridge slot of 0.5 m and a height 10.7 m at the gable peak. The height at the sides is 4.2 m. The internal room volume is about 25,000 m$^3$. The widely open long side walls are +17° rotated to the north-south axis to account for the main prevailing wind direction (i.e., south/south-west). In each gable wall, there is one gate and four doors with adjustable curtains. The building is designed for dairy cows in loose housing with littered lying cubicles and concrete walking alleys. In the measurement period, the open side walls were protected by nets. Air was introduced via adjustable curtains and four ceiling fans (Powerfoil X2.0, Big Ass Fans HQ, Lexington, KY, USA) with a diameter of 7.34 m (installed on a height of 5.6 m above the floor over the feeding alley) were operated climate-controlled under warm and low-wind conditions. The alleys were cleaned every 90 min by automatic scrapers. The cows were divided into four groups which differed in the average milk yield. Each group had cubicles with a deep bedding of chopped straw and chalk with a depth of around 0.2 m. A total mixed ration consisting of soy (24 %), oil-seed rape (19 %), maize (24 %), rye (23 %), and lupins (10 %) was fed twice a day (between 06:30 a.m. and 07:00 a.m. for all cows and between 09:00 a.m. and 12:00 a.m. in groups). The cows were milked in the neighboring milking parlour three times a day. Each milking cycle lasted around three hours, the first started at 06:00 a.m., the second at

02:00 p.m. and the third at 10:00 p.m., respectively. Each group left the barn for approximately 45 min during each milking cycle.

Farm B: The second data set originated from a naturally ventilated dairy building located in Brandenburg, Eastern Germany (approximately 56 km west of Berlin, 32 m above sea level) [26]. The building's total floor area had the dimensions 38.88 m × 17.65 m. The asymmetric roof is closed and has a height of 6.2 m at the gable peak. Two window arrays are included for additional lighting. The height at the sides is 3.6 m. The internal room volume is about 4500 m$^3$. The long side walls are +75° rotated to the north-south axis to account for the main prevailing wind direction (i.e., south-west) and the complex surrounding building structure. While the southern gable wall is widely closed, the northern gable wall has two large gates. About half of the eastern side wall is closed, while the western side wall has a reduced opening height of about 1.5 m. The building is designed for dairy cows in loose housing with littered lying cubicles and mainly concrete walking alleys, except for a small area in front of the automated milking system (Lely Astronaut A4, Maassluis, The Netherlands) that is equipped with slatted floor. With the automatic milking system embedded in the building, cows could go for milking any time of the day (no fixed milking periods). However, milking visits per day and cow are restricted, and a minimum of six hours between each milking event of a cow should be adhered to. In the measurement period, three axial fans, each with a diameter of 1.5 m, mounted on a height of approximately 3 m, were operated manually by the farmer. The alleys were cleaned hourly by an automatic scraper. The cubicles had a deep bedding of chopped straw and chalk with a depth of around 0.2 m. A total mixed ration consisting of maize (53 %), grass silage (23 %), rye (7 %), soy (6 %), oil-seed rape (7 %), lupins (2 %), and a mix of other supplements like feed lime or stock salt (2 %) was fed twice a day at 06:30 a.m. and 10:00 a.m. (remaining feed was moved towards the feeding places at 09:00 a.m., 12:00 a.m., 03:30 p.m., 05:00 p.m. and 08:00 p.m.). Additional concentrate was fed in the automated milking system based on individual days in milk and milk yield.

### 2.1.2. Measurement Setup

In the case of the larger building (farm A) four sampling lines representing outdoor concentrations and six sampling lines representing indoor concentrations, positioned at a distance of 4 m to 8 m to the walls, were used in this study [27]. The outdoor lines and five of the indoor lines were at a height of 3.2 m. The last indoor line was positioned in the middle of the barn below the ridge.

In the case of the smaller building (farm B), three sampling points representing outdoor concentrations and three sampling lines representing indoor concentration, were used. They were positioned with a minimum distance of approximately 3 m (indoor sampling lines) and 6 m (outdoor sampling points) from the walls. The inside lines were positioned at a height of 2.7 m, the outside points between 3 m and 5 m.

In all cases, the air was sucked through PTFE (i.e., Teflon) tubes with an inner diameter of 6 mm. Every 8 m to 10 m an orifice with a capillary trap was placed to ensure a uniform volumetric flow at each orifice. The measurement duration per line was 10 min and each line was accessed once per hour. At each line, concentrations of different gases, including carbon dioxide and methane, were measured in parallel from the air samples using one of two high resolution Fourier Transform Infrared (FTIR) spectrometer (Gasmet CX4000, Gasmet Technologies Inc., Karlsruhe, Germany). Carbon dioxide and methane concentrations were monitored on-farm from 1 November 2016 until 30 August 2017 on farm A and from 13 September 2017 until 28 February 2018 on farm B. In consequence, two datasets with 7276 (larger building) and 4056 (smaller building) hourly values of gas concentrations per sample line were available for this study.

Air temperature was measured on farm A with an EasyLog USB 2+ sensor (Lascar Electronics Ltd., Module House, Whiteparish, UK) close to the building at a distance of 5 m. On farm B it was measured at a distance of around 80 m with a weather station with an integrated PT100 Sensor. Outdoor temperature was considered, since it is more commonly available than indoor temperature while both are highly correlated (cf. [14] where for the barn on farm A indoor and outdoor temperature

were found to be correlated with a Pearson correlation coefficient larger than 0.98). The largest deviation between indoor and outdoor temperature can be expected for extreme cold conditions, where up to 5 °C higher indoor than outdoor temperatures were reported in the building on farm A in preceding studies during cold weather conditions [12,14]. On the other hand, local deviations of the indoor temperature from the average indoor temperature were also found to be approximately ±2 °C in both buildings in a preceding study [26]. The effect of these uncertainties in the reference temperature will be investigated in a sensitivity analysis described in Section 2.2.1.

Animal parameters such as number of cows in the barn, cow mass and milk yield were provided as herd averages by the administration of the farms as summarized in Table 1.

**Table 1.** Herd composition in the measurement period. Values are averages in terms of the median of the distribution over the measurement period.

|  | Farm A | Farm B |
|---|---|---|
| breed | Holstein-Friesian | Holstein-Friesian |
| number of cows | 355 lactating, no dry | 50 lactating, no dry |
| body mass in kg | 682 | 700 |
| milk yield in kg day$^{-1}$ | 39 | 34 |

### 2.1.3. Derivation of Hourly Emission Values

Hourly values of ventilation rate $Q$ were estimated using a mass balance of carbon dioxide (cf. Equation (1)) based on the animal heat production model of a cow at a temperature of 20 °C corrected by the average temperature of ambient air [21,28,29].

$$Q = \frac{N \cdot P_{CO_2}}{[CO_2]_{inside} - [CO_2]_{outside}} \tag{1}$$

Here $Q$ is the ventilation rate (i.e., the volume flow) in $[m^3\,h^{-1}]$, $N$ is the number of cows and $[CO_2]_{inside} - [CO_2]_{outside}$ is the carbon dioxide concentration difference between indoor and outdoor air in $[g\,m^3]$. $P_{CO_2}$ is the estimated $CO_2$ production per cow in $[g\,h^{-1}]$. It is a function of several animal parameters and the ambient temperature, further described in [21]. It has to be noted that our reference temperature here was considered as the outdoor air temperature. However, as described in Section 2.1.2 the offset between indoor and outdoor temperature is usually small. Moreover, the $CO_2$ production model is barely sensitive to the reference temperature (cf. [18]).

Indoor concentrations of $CH_4$ and $CO_2$ were estimated as an average of all sample lines inside the barn during the respective hour. For outdoor concentrations of $CH_4$ and $CO_2$ the sample line outside the barn with the lowest $CO_2$ concentration value during the respective hour was taken.

Methane emission values were calculated by multiplying ventilation rate $Q$ with the methane concentration difference between indoor and outdoor $[CH_4]_{inside} - [CH_4]_{outside}$. Subsequently, the estimated values were normalized to emissions per livestock unit $E_{CH4}$ $[g\,h^{-1}\,LU^{-1}]$, cf. Equation (2).

$$E_{CH4} = \frac{Q \cdot ([CH_4]_{inside} - [CH_4]_{outside}) \cdot LU}{N \cdot m} \tag{2}$$

Here one LU is the body mass equivalent of 500 kg, $N$ is the number of cows in the barn and $m$ is the averaged mass of cows in kg.

Finally, infinite and negative values were removed from the data sets. Such implausible values can occur when concentrations are close to the detection limits or concentration differences between inside and outside are small or inverted under low wind conditions. From the dataset of the larger barn approximately 8% of the data were excluded, while from the dataset of the smaller barn approximately 33% were excluded due to implausibility. In consequence, 6705 hourly emission values for the larger barn and 2718 for the smaller barn were included in the data analysis.

### 2.2. Data Analysis

We applied polynomial regression on both datasets as well as on different subsets of those to investigate the dynamics of the methane emissions in the course of the day dependent on the ambient temperature. We considered two types of subsets of the data.

Grouping 1: In the first version of the grouping each of the two datasets was subdivided into sets of distinct measurement hours (in terms of hour of the day). This grouping was considered in order to investigate the circadian pattern.

Grouping 2: In the second version of the grouping smaller subsets of four times seven days were considered in order to estimate the uncertainty that is associated with the selection of training data for the regression. The particular sample size was chosen as it is small enough to permit a sufficiently large number of different realizations of such subsets while it is expected to be large enough to provide robust estimations of the average emission over the whole dataset (cf. [25]). Two of the periods were randomly selected from warm months (i.e., monthly average temperature above 10 °C, namely May–October). Two were randomly selected from the remaining cold months. The reduced dataset was subsequently grouped into sets of distinct measurement hours as in the case of grouping 1.

#### 2.2.1. Sensitivity Analysis

First, we investigated the sensitivity of estimated regression coefficients on the considered reference temperature. To this end, 6 times 1000 normally distributed random numbers (with distribution means 0 °C, 1 °C, 2 °C, 3 °C, 4 °C and 5 °C) were added to the measured temperature before the regression. This analysis was performed using grouping 1.

Next, the sensitivity of the estimated regression coefficients on the selected training data was investigated using in total 5000 realizations of grouping 2. Out of those realizations sets were excluded from the analysis if they comprised less than four values for the respective hour or all values were below 5 °C or above 15 °C. In consequence, 2804 to 3114 realizations from the dataset of farm A and 681 to 799 realizations from the dataset of farm B remained per measurement hour for the regression analysis.

#### 2.2.2. Evaluation Scheme

Scatter plots were used to investigate the relationship between the methane emission and the outdoor temperature. The abscissa was associated with the temperature and the ordinate with the methane emissions. In addition, the datasets were subdivided into temperature classes of 5 °C width. Interquartile ranges of the observed emissions within each temperature class were calculated to visualize the variability that resulted from other explanatory variables (e.g., wind or barn management) besides the ambient temperature.

The coefficients of the fitted parabolas were plotted as dots for different hours of the day. The uncertainty that was associated with a particular selection of a training dataset (i.e., grouping 2) was indicated by plotting the standard deviation from all realizations as darkred (mean + standard deviation) and orange (mean - standard deviation) vertical bars.

In order to further evaluate the influence of the data acquisition strategy on the observed emission dynamics, the coefficients of the fitted parabolas per hour of the day were compared to the coefficients that were obtained in a preceding long-term study on farm A (cf. [14]).

Furthermore, the positions of the parabola vertices were calculated from the estimated regression coefficients from the complete datasets as well as when considering grouping 1.

## 3. Results

We monitored outdoor methane concentrations between 1.4 ppm and 32.8 ppm and indoor methane concentrations between 3.0 ppm and 80.3 ppm at farm A, and outdoor methane concentrations between 0.0 ppm and 15.0 ppm and indoor methane concentrations between 0.8 ppm and 51.4 ppm at farm B. At the same time, we monitored outdoor carbon dioxide concentrations between 353.0 ppm

and 693.0 ppm and indoor carbon dioxide concentrations between 393.0 ppm and 1448.8 ppm at farm A, and outdoor carbon dioxide concentrations between 0.0 ppm and 638.0 ppm and indoor carbon dioxide concentrations between 8.5 ppm and 1452.5 ppm at farm B. After removing nine measurement times with obviously unrealistically low carbon dioxide concentrations at farm B, the minimal observed methane concentration was 1.4 ppm outside and 4.8 ppm inside and the minimal observed carbon dioxide concentration was 368 ppm outside and 399 ppm inside. Further statistical parameters of the gas concentrations are listed in Table 2.

**Table 2.** Ranges and averages of the measured concentrations of $CH_4$ and $CO_2$ in ppm at the two farms during the respective measurement periods. Here, SD is the standard deviation, LQ is the lower quartile and UQ is the upper quartile. All statistics are calculated after outlier removal.

| Farm | Place | Gas | Mean | SD | Minimum | LQ | Median | UQ | Maximum |
|------|-------|-----|------|------|---------|------|--------|------|---------|
| A | inside | $CH_4$ | 16.4 | 10.2 | 3.0 | 8.6 | 13.2 | 22.3 | 80.3 |
| A | outside | $CH_4$ | 3.1 | 2.1 | 1.4 | 2.1 | 2.4 | 3.1 | 32.8 |
| A | inside | $CO_2$ | 598.0 | 145.6 | 393.0 | 485.0 | 551.2 | 683.8 | 1448.8 |
| A | outside | $CO_2$ | 409.2 | 30.7 | 353.0 | 391.0 | 405.0 | 421.0 | 693.0 |
| B | inside | $CH_4$ | 18.8 | 6.7 | 4.8 | 13.8 | 18.5 | 23.7 | 51.4 |
| B | outside | $CH_4$ | 3.3 | 1.2 | 1.4 | 2.5 | 3.1 | 3.9 | 15.0 |
| B | inside | $CO_2$ | 597.8 | 107.8 | 399.0 | 522.5 | 587.0 | 663.3 | 1452.5 |
| B | outside | $CO_2$ | 414.3 | 34.4 | 368.0 | 402.0 | 413.0 | 424.0 | 638.0 |

From these measured gas concentrations we derived hourly methane emission values as described in Section 2.1.3.

*3.1. Functional Shape and Variability*

The scatter plots of methane emission over ambient outdoor temperature show a considerable increase of emission values for temperatures above 15 °C or below 5 °C (about $0.5\,\mathrm{g\,LU^{-1}\,h^{-1}}$ per 1 °C) for both measurement sites as shown in Figure 2 in the first column. The average methane emission value for farm A was found to be approximately $11.5\,\mathrm{g\,LU^{-1}\,h^{-1}}$, while that of farm B was approximately $14.2\,\mathrm{g\,LU^{-1}\,h^{-1}}$. In both cases, a considerable variability of the emission values can be observed. The lower quartiles of the distribution of the complete datasets were calculated as $9.3\,\mathrm{g\,LU^{-1}\,h^{-1}}$ for farm A and $11.7\,\mathrm{g\,LU^{-1}\,h^{-1}}$ for farm B. The respective upper quartiles were $13.7\,\mathrm{g\,LU^{-1}\,h^{-1}}$ for farm A and $16.5\,\mathrm{g\,LU^{-1}\,h^{-1}}$ for farm B. This results in an interquartile range of $4.4\,\mathrm{g\,LU^{-1}\,h^{-1}}$ for the larger and $4.8\,\mathrm{g\,LU^{-1}\,h^{-1}}$ for the smaller measurement site. The interquartile range was rather homogeneous among the different temperature classes (as illustrated in the second column of Figure 2) and indicates that there is a constantly large variability associated with other factors besides temperature (e.g., wind conditions or herd management).

Over the day the average methane emission value varied (averaged over all observed outdoor temperatures) between approximately $8\,\mathrm{g\,LU^{-1}\,h^{-1}}$ and $15\,\mathrm{g\,LU^{-1}\,h^{-1}}$ on farm A and between approximately $10.5\,\mathrm{g\,LU^{-1}\,h^{-1}}$ and $16.5\,\mathrm{g\,LU^{-1}\,h^{-1}}$ on farm B as shown in Figure 3 in the upper left panel.

Lower average methane emissions (in terms of the median of the distribution) were observed during the night and highest values in the afternoon. The opposite behavior was observed for the differences between outside and inside methane concentration where the minimum was observed around midday and high concentration differences were observed during the night (results not shown). In contrast, the diurnal rhythm of the emissions is well inline with the observed diurnal pattern of the air exchange rate on both farms as shown in Figure 3 in the lower left panel. Aggregated over the whole day typically lower air exchange rates (approximately $41\,\mathrm{h^{-1}}$ vs. $50\,\mathrm{h^{-1}}$) as well as higher concentration differences (approximately 15 ppm vs. 13 ppm) and emission values (approximately $14\,\mathrm{g\,LU^{-1}\,h^{-1}}$ vs. approximately $11\,\mathrm{g\,LU^{-1}\,h^{-1}}$) were observed on farm B compared to farm A.

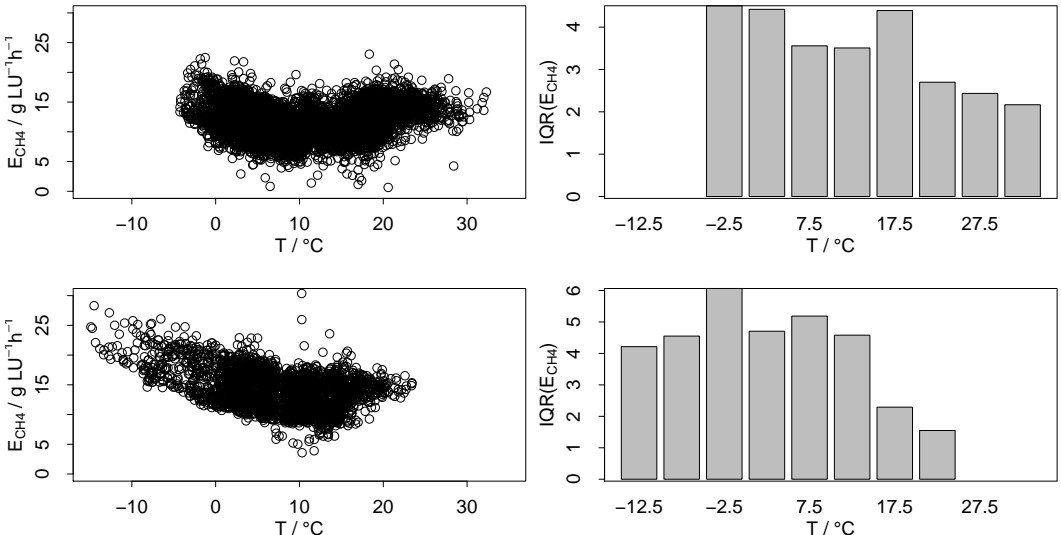

**Figure 2.** The upper panels show the distribution of methane emissions from the building on farm A, the lower from the building on farm B. (**Left**): Scatter plots of methane emission values over the outdoor temperature. Despite the variability, there are clear increasing trends in the emission values if the temperature considerably decreases from approximately 10 °C and if it considerably increases from approximately 10 °C. (**Right**): Interquartile ranges (IQR) of the distribution of methane emission values within classes of outdoor temperature (each 5 °C width). In most cases the IQR was found to be between 3 g LU$^{-1}$ h$^{-1}$ and 5 g LU$^{-1}$ h$^{-1}$. This indicates that there is a large variability that cannot be explained by the ambient temperature alone.

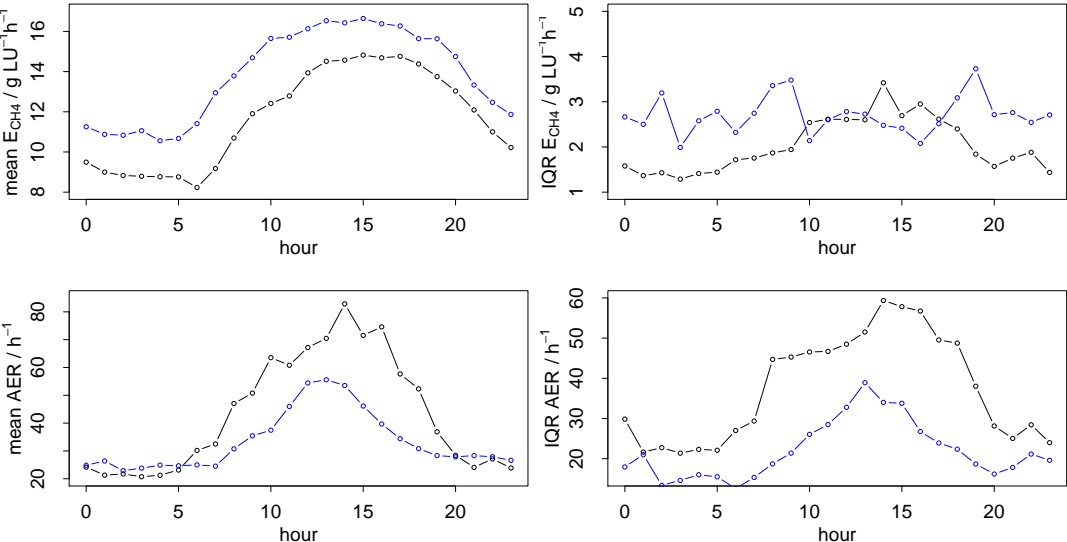

**Figure 3.** Distribution of methane emission values ($E_{CH4}$) and air exchange rates (AER, i.e., volume flow per total volume) during the day at farm A (black) and farm B (blue). The (**left**) panels show the median values. The (**right**) panels shows the associated interquartile ranges (IQR).

The respective interquartile ranges of the emission values (see Figure 3 in the right panels) showed a similar variability as among the temperature classes. In the morning and during the night a slightly larger variability of the emission values was observed on farm B as compared to farm A. This is different from the pattern observed for the interquartile ranges of the air exchange rates, where particularly on farm A during the day large variability was observed. Highest variability in the air exchange rates was

observed in the early afternoon around the time where the highest average air exchange rates have been found.

Considering the scatter plots from both measurement sites the parabolic shape of the emission values over temperature persists when considering subsets of different hours of the day (see Figure 4).

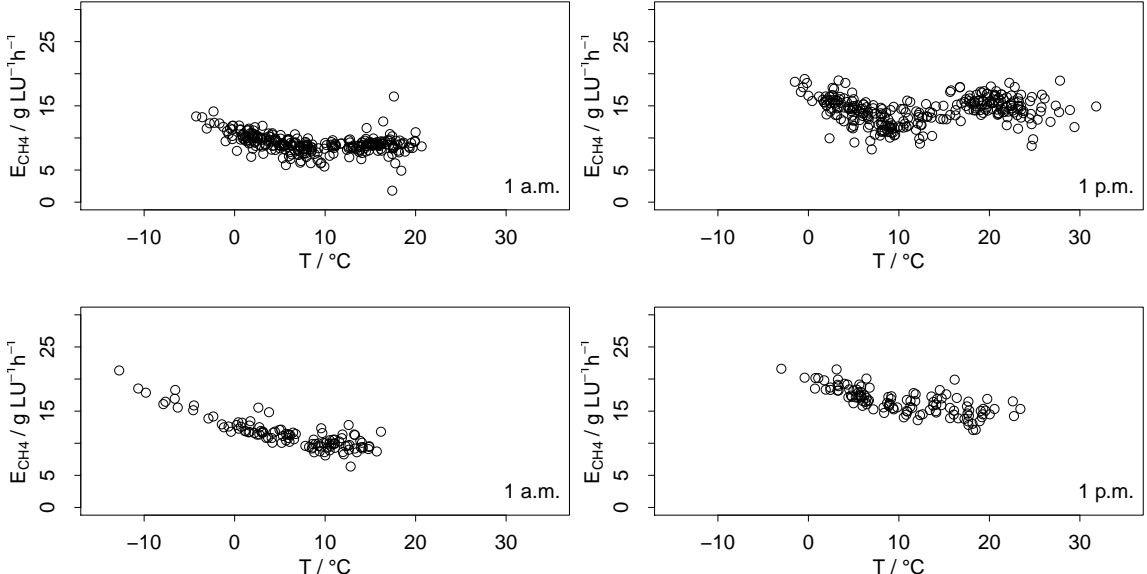

**Figure 4.** Scatter plots of methane emission over temperature for two different hours of the day, namely in the night at 1 a.m. (**left**) and in the afternoon at 1 p.m. (**right**). The upper panels show data from farm A and the lower panels from farm B.

### 3.2. Hourly Coefficients

Since we observed slightly different parabolic shapes when considering subsets of different hours of the day (cf. Figure 4), we fitted a polynomial of second order for each hour of the day and compared the obtained coefficients to those found in an earlier study of the same building on farm A. As shown in Figure 5 the coefficients that were estimated from the old and the current training dataset were rather similar in terms of absolute differences (although due to the small values relative differences of more than 100 % were observed in some cases). While the linear coefficient was slightly higher in the old dataset, the quadratic coefficient was slightly lower. When the temperature uncertainty (resulting from the indoor variability of the air temperature and the typically observed offset between indoor and outdoor air temperature) was considered in the fitting procedure, the values of the linear and the quadratic coefficients converged even more towards the values estimated in the earlier study which used indoor temperature as a reference.

In contrast to the linear and the quadratic coefficients of the polynomial, which show no distinct circadian pattern, we found an approximately sinusoidal behavior of the constant coefficient over the day. The sine had an amplitude of approximately $4\,\mathrm{g\,LU^{-1}\,h^{-1}}$, an offset on the ordinate of approximately $13\,\mathrm{g\,LU^{-1}\,h^{-1}}$ and an offset on the abscissa of approximately $8\,\mathrm{h}$. The impact of the temperature uncertainty on the constant coefficient of the polynomial was negligible.

In order to further evaluate the robustness of the fitted coefficients, we repeated the estimation of the coefficients on small subsets of the dataset as described in Section 2.2. While the consideration of small subsets led to a considerable spreading of the estimated coefficient value, for the constant coefficient the average over all realizations was very close to the estimation based on the complete dataset as shown in Figure 5. Larger deviations occurred for the linear and particularly the quadratic coefficients indicating that the fit of these coefficients is more sensitive to the selection of the training data.

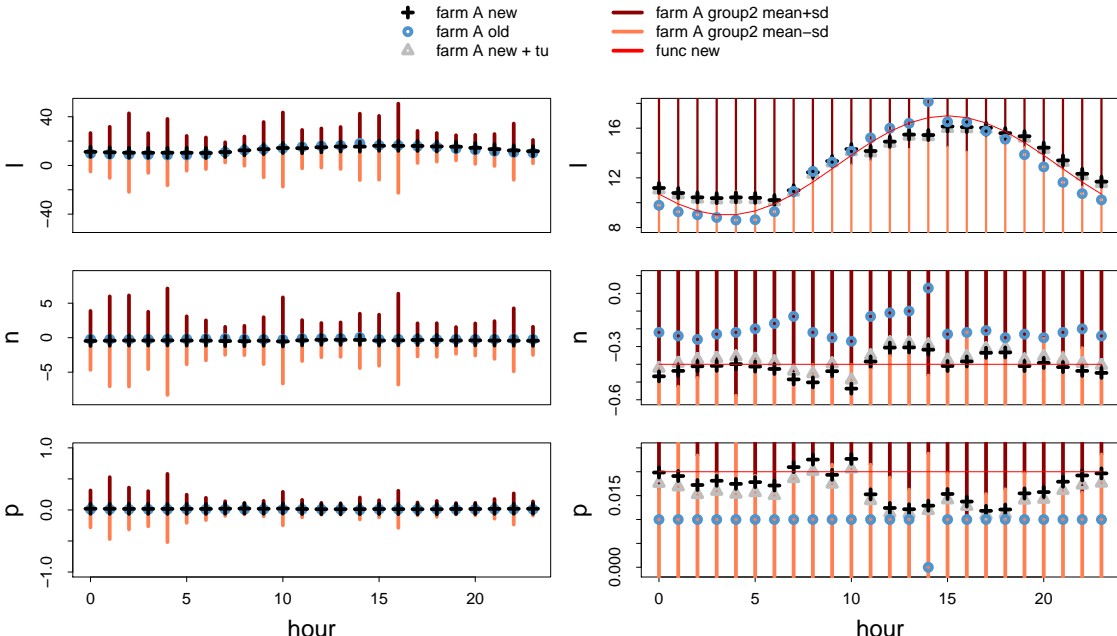

**Figure 5.** Coefficients l, n and p obtained when fitting the polynomial $E_{CH4} = l + n \cdot T + p \cdot T^2$ to the dataset of farm A. Referring to the notation in [14] the coefficients are denoted l (constant term), n (linear term) and p (quadratic term). The values of the coefficients obtained from grouping 1 with the actually measured temperature values are denoted "farm A new" and the associated averages of all realizations with temperature uncertainty are denoted "farm A new + tu". In addition, the spread among the realizations obtained using grouping 2 is indicated by the vertical lines ("farm A group 2 mean + sd" and "farm A group 2 mean − sd", where sd refers to the standard deviation). For comparison, the coefficients obtained in [14] for the same farm are plotted ("farm A old"). The right panels are zoomed versions of the left panels. In addition, on the right the assumed dynamics of the parameters in the course of the day are indicated by "func new".

We repeated the same analysis with the smaller dataset that was associated with farm B (see Figure 6) to investigate the impact of the measurement site. In this case, only about one forth of the data amount relative to the first site was available. We found that the variability when considering only four-week subsets of the data to estimate the polynomial was considerably higher than in the case of farm A in Figure 5. Using the whole dataset, however, the dynamics of the coefficients were very similar. The linear and the quadratic coefficient fitted for the measurement site on farm A were even more similar to those fitted for farm B than to the coefficients estimated in the earlier study for farm A. The linear coefficient was on average slightly lower for farm B than for farm A ($-0.41\,\mathrm{g\,LU^{-1}\,h^{-1}\,{}^\circ C^{-1}}$ versus $-0.48\,\mathrm{g\,LU^{-1}\,h^{-1}\,{}^\circ C^{-1}}$). The quadratic coefficient was on average about $0.015\,\mathrm{g\,LU^{-1}\,h^{-1}\,{}^\circ C^{-2}}$ on farm B (almost the same as on farm A with $0.017\,\mathrm{g\,LU^{-1}\,h^{-1}\,{}^\circ C^{-2}}$). The constant coefficient for farm B also showed a distinct sinusoidal behavior. The estimated amplitude of this sine was with approximately $6\,\mathrm{g\,LU^{-1}\,h^{-1}}$ slightly larger than for the other measurement site. Moreover, the offset on the ordinate was with approximately $16\,\mathrm{g\,LU^{-1}\,h^{-1}}$ slightly larger, while the offset on the abscissa was again approximately 8 h. This means that the daily average as well as the daily maximum emissions per livestock unit can be expected to be slightly higher on farm B compared to farm A. The maxima were, however, observed at similar time frames in both buildings.

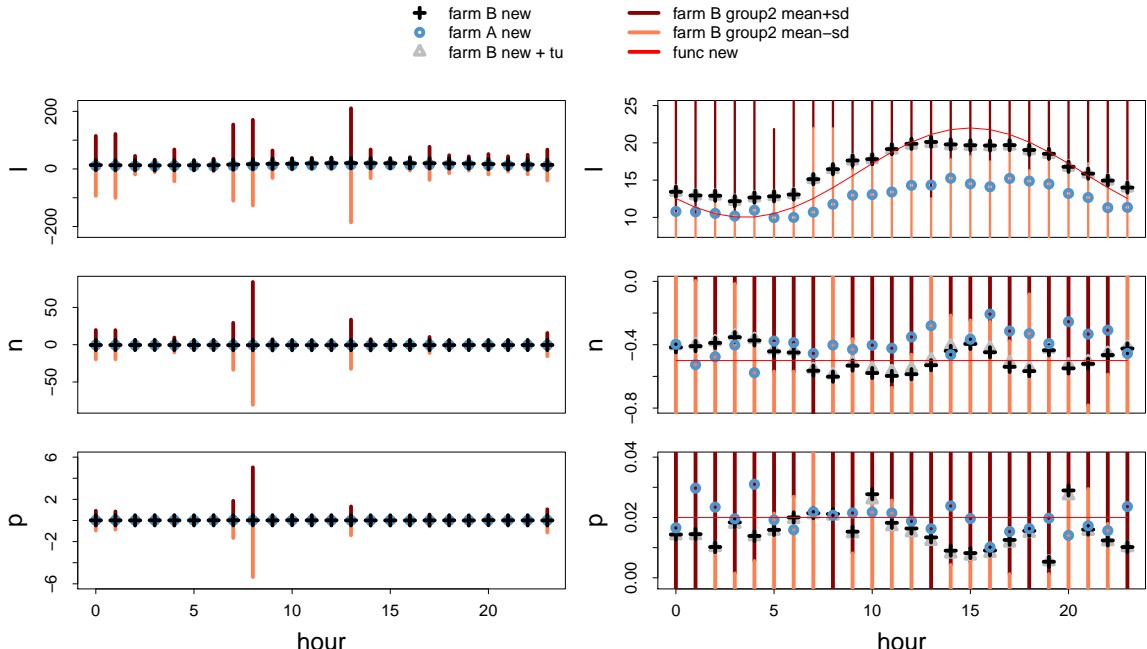

**Figure 6.** Coefficients l, n and p obtained when fitting the polynomial $E_{CH4} = l + n \cdot T + p \cdot T^2$ to the dataset of farm B (cf. Figure 5). The values obtained from grouping 1 with the actually measured temperature values are denoted "farm B new" and the associated averages of all realizations with temperature uncertainty are denoted "farm B new + tu". In addition, the spread among the realizations obtained using grouping 2 is indicated by the vertical lines ("farm B group 2 mean + sd" and "farm B group 2 mean − sd", where sd refers to the standard deviation). For comparison, the coefficients obtained for farm A from grouping 1 with the actually measured temperature are plotted (denoted "farm A new"). The right panels are zoomed versions of the left panels. In addition, on the right the assumed dynamics of the parameters in the course of the day are indicated by "func new".

### 3.3. Comparison of Vertex Position

The positions of the respective parabola vertices were deduced from the polynomial regression coefficients and are compared in Figure 7. Considering coefficients that were obtained from the whole data set as well as from the hourly data in grouping 1, we observed a substantial variability of the expected minimal emissions over the day. The average over the day was, however, comparable to the minimal emissions deduced from the entire dataset without grouping. Furthermore, it can be seen that the distribution of the values obtained from the grouping 1 for farm B was much broader where some of the fits led to unreasonably high optimal temperatures (probably due to an insufficient amount of usable data for the fitting procedure). Reasonable optimal temperatures between 10 °C and 15 °C were obtained for all hours for farm A with almost ten months of measurements. Comparing the vertices derived from the two complete datasets it can be seen that the optimal temperatures as well as the minimal emission value were slightly higher for farm B.

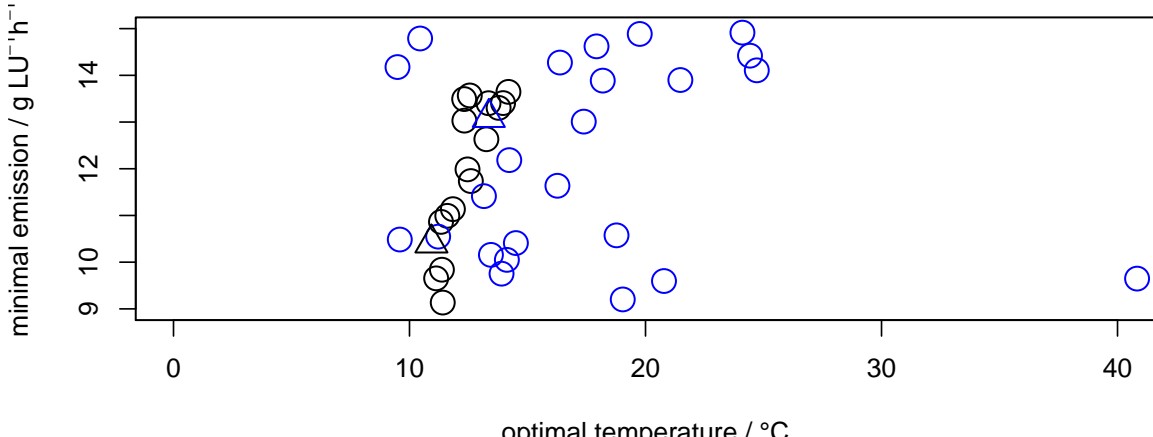

**Figure 7.** Temperature optimum and minimal emission estimated from regression of grouping 1 and from regression of polynomial of second order for the whole dataset for farm A (black) and farm B (blue).

## 4. Discussion

### 4.1. Temperature Optimum for Methane Mitigation

Our analysis has shown that the parabolic shape of the temperature dependence of methane emissions, which has been found in a preceding study by [14], is also detectable with a different measurement setup (sampling lines instead of sampling points and infrared spectroscopy instead of photo acoustic analyzer). Further, we have shown that similar dynamics can be found at a different measurement site with a comparable dairy cow management (performance and feeding), floor type and manure management. Following our aim to analyse the parabolic dependence of methane emissions on the air temperature and to analyse the variation in shape of the parabola in the course of the day, we did not include the influence of feed composition and milk yield on the actual amount of methane emissions. Both herds were constantly high yielding dairy cows with only little variation in milk yield. They were fed a totally mixed ratio of high quality. The observed variation in methane emissions was not caused by variations in milk yield or feed quality.

Despite a large uncertainty in the independent and dependent variables the shape of the curve has been found to be robust. While for housing systems with manure storage this dynamics can be expected to be masked by superimposed dynamics of methane emissions from the manure, the identified parabolic temperature dependence in our study is very likely associated with the ruminants' metabolisms. Thermal discomfort as a potential cause of elevated methane emissions is related to stress which has an effect on the digestive system. For example, reference [30] reported remarkable shifts in glucose and lipid metabolism in hyperthermic cows. Lipid metabolism on the other hand is known to occur predominantly in the rumen, where lipids from the diet enter the rumen and are hydrolyzed into their constituent components [31]. Further changes in the levels of fatty acids and glucose have been found during hypothermia indicating also changes in the metabolism in dependence of the ambient temperature [32]. Reference [32] further reported that exposure to low temperature increases the voluntary feed intake but decreases the digestibility, while the opposite is true for exposure to high temperature. As methanogenesis in the rumen is related to the amount of feed intake and the digestibility, this immediately suggests that there must be an optimal temperature where the direct methane release from the ruminant is minimal. This minimum has been found to lie approximately between 10 °C and 15 °C for the measurement sites in our study, which is well in line with previous findings for farm A where minimal emissions were reported around 10 °C (cf. [14]). In our current study, the temperature optimum and the emission minimum were found to be slightly higher on farm B than on farm A (i.e., approximately 2.5 °C and 2.7 g $LU^{-1} h^{-1}$ which is

approximately 22% of the average emission value of the two locations). The observed differences in the optimal temperature can be partly explained by the fact that the outdoor temperature in summer was in the long-term average about 1 °C higher in the region of farm B, which may have let to physiological adaptation of the animals. Another possible influencing factor could be a slightly different herd composition as high-yielding cows are producing more metabolic heat than lower-yielding ones and thus their thermal optimum is at lower temperatures (cf. [33] who estimated the critical temperature to be approximately 12 °C lower for cows with a milk yield of 30 kg per day compared to dry cows). An other part of the temperature offset could be explained by the variability that was particularly observed in the dataset of farm B and led to unreasonable estimates for the optimal temperature for some hours when considering the reduced datasets of hourly grouped values.

This variability and the resulting uncertainty could explain also a part of the offset of the minimal emission value, since high fluctuations at farm B were observed particularly for temperatures around 10 °C pushing the estimated vertex position towards higher emission values (cf. Figure 2).

In addition, a part of the offset in the emission values on both farms might be also associated with the accuracy of the measurement device and the gas sampling setups.

Besides this uncertainty, slightly different feeding strategies (e.g., more maize on farm B) and herd compositions (e.g., slightly higher average body mass on farm B) can explain a part of the observed offset as the amount of methane release from cattle depends on their dry matter intake, milk yield and on the metabolic weight of a cow [23].

Moreover, the slightly different diurnal patterns of the air exchange rate that were observed at the two measurement sites could explain a part of the offset between the measurement sites than considering daily emission values. While on farm A average air exchange rates of around $50\,h^{-1}$ were found, the average air exchange rates on farm B were considerably lower at around $41\,h^{-1}$. This might lead to an accumulation of methane in the building during the night, which is in line with the observation that particularly high concentration differences between outside and inside were observed during the night. The different magnitudes of air exchange rates on both farms can thus result in different amplitudes of the diurnal variation of methane emissions on both farms as found in our study.

## 4.2. Circadian Rhythm as Mitigation Option

The coefficients that were derived from the polynomial regression of second order showed characteristic dynamics over the day which were similar for both measurement sites. The coefficient of the linear and the quadratic term were rather constant throughout the day, where the coefficient of the linear term was one order of magnitude larger than the coefficient of the quadratic term. This means that, considering a temperature optimum of around 10 °C both terms will contribute nearly equally to the description of the total methane emission value. The coefficient of the constant term was found to be approximately two orders of magnitude higher than the coefficient of the linear term and three orders of magnitude higher than the coefficient of the quadratic term. Thus, this constant term, which showed a distinct dependence on the hour of the day, is clearly governing the overall dynamics of methane emissions. The circadian pattern that has been found for the coefficient of the constant term is well in line with previous findings from [12] who found the diurnal methane emissions peaking at 2 p.m. on farm A, approximately 2 h after the second feed supply. On farm B, we observed that the methane emission value went up a little bit earlier than on farm A which could be related to the fact that the first feed provision on farm A is administered while the first milking cycle is still on-going. In contrast on farm B cows use the automatic milking system and might be timely in place for feed intake. In contrast to the pattern that we observed, refs. [23,34] reported rather a bimodal shape of the emission curve with peaks around 9 a.m. and 5 p.m., both approximately 1 h after feed supply [23]. This apparent discrepancy in the dynamics can, however, be explained with different feeding times and particularly different duration between feeding events. According to [35] the diurnal pattern of enteric methane emissions from dairy cows can be modeled knowing the final

asymptotic accumulated enteric methane emissions for feeding and the ruminal passage rates as well as two shape parameters [35]. Following this model short intervals between feeding events will lead to one pronounced emission peak after the last feeding as the enteric methane emission generated by the residual substrate in the rumen is still significant at the time when the next feeding starts. This describes well the dynamics which we observed for the coefficient of the constant term of our polynomial regression.

As the diurnal pattern of the methane emissions is mainly governed by the feeding time and the duration between feedings, but further modulated by the ambient temperature, feeding strategies could be explored as a mitigation measure not only in terms of feed composition, but also in terms of timing. Feed intake predominately during colder periods of the day when a better digestibility of the feed can be expected might have a positive effect on the production efficiency (i.e., more milk yield and less methane from the same amount of feed intake). As feed intake only during parts of the day, however, might be problematic in the context of nutrition of high-yielding cows, keeping indoor temperature constantly around 10 °C would be a promising option for methane emission mitigation. However, under which conditions the temperature effect can be significant, must be further explored in a separate detailed study. Moreover, the potential interdependence with other husbandry-related factors must be investigated in detail.

## 5. Conclusions

We considered emission data from two measurement sites of different size with littered lying cubicles and frequently cleaned mainly solid walking areas. In both buildings high-yielding Holstein-Friesian dairy cows were housed. The smaller building had about one fifth of the volume of the larger one and housed about one seventh of the number of cows. Both measurement sites showed slightly different air exchange rate and emission pattern over the day. The differences in the emission pattern are likely a consequence of the differences in air exchange rates together with slight differences in the feeding times and the different milking systems. The dependency of the emission value on the ambient temperature was very similar on both farms. The observed functional shape was also comparable to the findings of a prior study on the larger measurement site with a different measurement setup. This supports our initial hypothesis that the methane emissions from naturally ventilated dairy barns with concrete floor show a parabolic dependence on the air temperature independent of the gas sampling and averaging strategy. In consequence, we conclude that empirical modeling of methane emissions from this type of buildings with a polynomial of second order for the independent variable air temperature can increase the accuracy of predicted long-term emission values for regions with pronounced seasonal temperature fluctuations.

We further found that when applying a respective polynomial regression of second order, the coefficient of the constant term reflects the diurnal pattern of methane emissions that was reported in earlier studies and is associated with the feed intake. In contrast, the coefficients of the linear and quadratic term were found to be rather stable over the day. In consequence, the position of the fitted vertex of the parabola moves slightly in the course of the day. However, we could identify a herd specific emission optimum at ambient temperatures around 10 °C to 15 °C with the best compromise of feed intake and digestibility for minimal methane emissions. As this coincides with the welfare temperature of dairy cows, we conclude that temperature regulation of dairy cow buildings with concrete floor should be considered and further investigated as an emission mitigation measure.

**Author Contributions:** S.H., D.W., D.J., C.A. made major contributions to the conceptualization of the data collection and the methodology selection in this study. B.A. and T.A. contributed to the conceptualization of the study framework. D.W., D.J. and C.A. carried out data preprocessing. S.H. implemented the analysis algorithm, carried out the validation and visualization setting. All authors contributed to the interpretation. S.H., D.W., D.J. and C.A. took care of data curation. S.H. wrote the original draft. All authors contributed in the reviewing and editing of the draft. All authors have read and agree to the published version of the manuscript.

**Funding:** This research received no external funding.

**Acknowledgments:** We thank Ulrich Stollberg and Andreas Reinhard, technicians at ATB, for technical support during the measurements, Anke Römer, Bernd Losand and Christiane Hansen from the Landesforschungsanstalt für Landwirtschaft und Fischerei Mecklenburg-Vorpommern (LFA-MV) as well as the staff of Gut Dummerstorf and LVAT Groß Kreutz for the comprehensive provision of climate and animal data.

**Conflicts of Interest:** The authors declare no conflict of interest. The funders had no role in the design of the study; in the collection, analyses, or interpretation of data; in the writing of the manuscript, or in the decision to publish the results.

## Abbreviations

The following abbreviations are used in this manuscript:

FTIR    Fourier Transform Infrared
LU      Livestock Unit (500 g body mass equivalent)

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
