# Peer review of "Methane Emission Characteristics of Naturally Ventilated Cattle Buildings"

_sustainability, doi:10.3390/su12104314_

Round 1
Reviewer 1 Report
The issues contained in the work entitled: „Methane emission characteristics of naturally ventilated cattle buildings” are current and important in the context of the quantification of the greenhouse gases sources and its emissions to the atmosphere and the search for ways to reduce them. In my opinion, the paper was developed properly. The research problem was well outlined and research hypotheses were proposed, which were verified. Below are some comments.
Material and Methods
- Lines 67 and 77: The periods of studies should be detailed (at least half a year / about a year?). In the farm A, the study were conducted from 01-11-2016 to 30-08-2017 (10 months), and in the farm B from 13-09-2017 to 28-02-2018 (about 5 and a half months).
- Lines 85 and 104: These are dimensions, not areas.
- Please, Consistently describe the milking system in the farm B, all the more so as it was shown later in the paper that this could have had an impact on the methane emission pattern.
- Equation 2: Methane emissions have been marked with the symbol En? In the figures are ECH4. Please unify.
Results
- I am aware that with such a large amount of data it is not possible to present it fully in the paper, but please provide at least the ranges/averages of methane and carbon dioxide concentration obtained at the sampling sites.
- 3: Please explain the abbreviation AER, how this indicator was determined?
- Figures 5 and 6: Please explain the abbreviations p, l, n.
Discussion
In the section "Discussion" the analysis of the impact of feed quality on the magnitude of CH4 production and its dynamics in farms was basically omitted. The composition of feed used in both farms differs quite significantly. This issue should be discussed in detail (if it possible).
Author Response
Please find our reply in the attached file.

Reviewer 2 Report
Dear authors,
In the paper entitled "Methane emission characteristics of naturally ventilated cattle buildings" you showed an interesting research on the factors affecting methane emissions from naturally ventilated free-stall barns with littered cubicles and a concrete floor in corridors, in the context of reducing methane emissions from ruminants.
The work is well organized and comprehensively described. I rate this work highly in every respect. Nevertheless, there are a few things to improve of the manuscript, primarily: some typos, units, improvement of some words, supplementing some figures and some technical things.
All comments and corrections were made to the text of the manuscript.

Author Response
Please find our reply to the review in the attached file.
